# COVID-19-Triggered Acute Liver Failure and Rhabdomyolysis: A Case Report and Review of the Literature

**DOI:** 10.3390/v15071445

**Published:** 2023-06-27

**Authors:** Yukako Matsuki, Takaaki Sugihara, Takuya Kihara, Tatsuru Kawakami, Tsuyoshi Kitaura, Tomoaki Takata, Takakazu Nagahara, Kai Fujita, Masayuki Hirai, Masaru Kato, Koichiro Kawaguchi, Hajime Isomoto

**Affiliations:** 1Division of Gastroenterology and Nephrology, Department of Multidisciplinary Internal Medicine, Faculty of Medicine, Tottori University, Yonago 683-8504, Japan; matsukiy@tottori-u.ac.jp (Y.M.); t.kihara@tottori-u.ac.jp (T.K.); t-takata@tottori-u.ac.jp (T.T.); t.nagahara@tottori-u.ac.jp (T.N.); koichiro@tottori-u.ac.jp (K.K.); isomoto@tottori-u.ac.jp (H.I.); 2Division of Infectious Diseases, Faculty of Medicine, Tottori University, Yonago 683-8504, Japan; m1234tk@gmail.com (T.K.); kitaurat@tottori-u.ac.jp (T.K.); 3Division of Medicine and Clinical Science, Department of Cardiovascular Medicine and Endocrinology and Metabolism, School of Medicine, Faculty of Medicine, Tottori University, Yonago 683-8504, Japan; kfujita@tottori-u.ac.jp (K.F.); m-hirai@tottori-u.ac.jp (M.H.); kato3@tottori-u.ac.jp (M.K.)

**Keywords:** acute liver failure, viral-induced acute liver failure, rhabdomyolysis, viral-induced rhabdomyolysis, COVID-19, SARS-CoV-2 infection

## Abstract

COVID-19 is primarily known for its respiratory tract involvement, often leading to severe pneumonia and exacerbation of underlying diseases. However, emerging evidence suggests that COVID-19 can result in multiorgan failure, affecting organs beyond the respiratory system. We present the case of a 62-year-old male with COVID-19 who developed acute liver failure (ALF) and rhabdomyolysis in the absence of respiratory failure. Initially, the patient presented with significantly elevated aspartate transaminase (5398 U/L) and alanine transaminase (2197 U/L) levels. Furthermore, a prolonged prothrombin time international normalized ratio (INR) of 2.33 indicated the diagnosis of ALF without hepatic coma, according to Japanese diagnostic criteria. The patient also exhibited elevated creatine kinase (9498 U/L) and a mild increase in creatinine (1.25 mg/dL) levels, but both values improved with intravenous fluid support and molnupiravir administration. To our knowledge, this is the first reported case presenting with both ALF and rhabdomyolysis associated with COVID-19. In addition, we review the existing literature to summarize previously reported cases of ALF triggered by SARS-CoV-2. This case report underscores the significance of recognizing COVID-19 as a significant contributing factor in the development of multiorgan failure. Furthermore, it suggests that COVID-19 can lead to severe illness, irrespective of the absence of respiratory failure.

## 1. Introduction

On 11 March 2020, the World Health Organization (WHO) officially declared the severe acute respiratory syndrome coronavirus 2 (SARS-CoV-2) as a global pandemic [1]. As of the end of March 2023, the WHO received reports of over 761 million confirmed cases of coronavirus disease 2019 (COVID-19) worldwide, with an unfortunate toll of nearly 7 million deaths [2]. COVID-19 is known to be a systemic disease impacting various organs and bodily functions. Alongside respiratory failure, hospitalized patients have shown manifestations of acute kidney injury (9%), liver dysfunction (19%), bleeding and coagulation dysfunction (10–25%), and septic shock (6%) [3]. Recent reports suggest a potential association between COVID-19 and rhabdomyolysis [4,5,6,7,8,9,10,11,12,13,14,15,16,17,18,19,20,21,22,23]. In this context, we encountered a case involving a patient with COVID-19 who exhibited both acute liver failure (ALF) and rhabdomyolysis, despite the absence of respiratory failure. This serves as an example of the diverse range of complications that can arise from SARS-CoV-2 infection. In Japanese diagnostic criteria, patients showing prothrombin time values of 40% or less of the standardized values or international normalized ratios (INR) of 1.5 or more caused by severe liver damage within 8 weeks of the onset of the symptoms are diagnosed as having ALF [24]. Rhabdomyolysis is diagnosed by elevations in serum creatine kinase (CK), and while there is no established serum level cut-off, many clinicians use five times the upper limit of normal (~1000 U/l) [25].

## 2. Case Presentation

A lean (BMI 13 kg/m^2^) 62-year-old Japanese man without known chronic disease was transferred to the emergency department of our hospital due to severe anemia and liver dysfunction. He was confused (Glasgow Coma Scale: E4V4M6), and his laboratory tests showed extremely high CK values. He was a nonsmoker and nonalcoholic. He sometimes used Etizolam for panic disorder. His family history was unknown for any neuromuscular disorders or myopathies. Due to a previous experience of anaphylaxis following influenza vaccination, he had never had a vaccination against SARS-CoV-2. He had contacted a person with COVID-19 three days before and complained of generalized fatigue, decreased appetite, and fever a day before the transfer. He did not take any antipyretic analgesics. His initial vital signs showed a blood pressure of 138/63 mm Hg, heart rate of 108 beats/minute, a temperature of 36.9 °C, a respiratory rate of 21 breaths/minute, and oxygen saturation of 98% on room air. A shock status or a hypotensive event was not documented throughout the transference. On physical examination, he had no tenderness over the major muscle group and no lymphadenopathy, and his liver and spleen were not palpable. No remarkable asterixis was observed. His initial laboratory tests are shown in Table 1.

The patient presented with a case of severe anemia, characterized by a significant decrease in hemoglobin levels. A Focused Assessment with Sonography for Trauma (FAST) scan did not reveal any intra-abdominal echo-free space, ruling out the presence of abdominal bleeding. Further investigations, including nasogastric tube insertion and digital rectal examination, showed no signs of gastrointestinal bleeding. Chest-computed tomography (CT) imaging demonstrated bronchopneumonia in the left upper lobe of the lung (Figure 1A). Contrast-enhanced CT imaging demonstrated the presence of a periportal collar sign, suggesting the presence of acute hepatitis (Figure 1B). Hepatic congestion was not remarkable, and no indications of ischemic changes were observed in either the liver or skeletal muscles. Gastroduodenal endoscopic examination unveiled a scar (S1 stage of Sakita–Miwa Classification) from a peptic ulcer located at the lesser curvature of the stomach (Figure 1C). However, the patient did not report any prior melena. The background mucosa displayed signs of chronic gastritis. The patient tested positive for Helicobacter pylori antibodies. Fecal occult blood tests were negative on two occasions. He declined to have a colonoscopy. The severe anemia was attributed to a peptic ulcer.

A chest radiograph exhibited a cardiothoracic ratio (CTR) of 67.6% (Figure 2A), while brain natriuretic peptide (BNP) levels measured 2937 pg/mL. Electrocardiographic monitoring findings did not exhibit abnormalities such as ST elevation or negative T waves. Echocardiography demonstrated an enlarged left ventricle (LVDd 53.5 mm, LVDs 42.0 mm) and a ventricular ejection fraction (vEF) ranging from 20% to 25%. A severe reduction in wall motion was observed throughout the myocardium. The characteristic features of “takotsubo cardiomyopathy,” including apical ballooning, symmetrical regional abnormalities in the form of a circumferential pattern, and left ventricular outflow tract obstruction, were absent in the observed case. Notably, there were no indications of cardiac valvular disease. Moreover, the CK–MB values (64 U/L) displayed a relatively modest elevation compared to the high CK values, which deviated from the expected patterns typically observed in cases of myocarditis. These observations significantly reduced the likelihood of myocarditis as a plausible diagnosis. Furthermore, despite calcification in the coronary arteries, the absence of wall-thinning suggested that acute coronary syndrome was an unlikely explanation.

While the potential for type 2 myocardial infarction resulting from anemia was considered, the cardiologists ultimately determined that the patient’s heart failure was secondary to stress-induced myocardial injury, particularly in severe conditions such as anemia, liver failure, and rhabdomyolysis. Consequently, no specific therapeutic interventions to address heart failure were administered, and the patient received red blood cell transfusions to manage the underlying anemia.

Transaminases were extremely elevated, and the prothrombin time international normalized ratio (PT-INR) was 2.33. The patient was also diagnosed with ALF without hepatic coma, according to the Japanese criteria for ALF [24]. Finally, no other causes of ALF were indicated, except for SARS-CoV-2 infection. Immediately, erythrocyte transfusion and cryoprecipitate were supplied. Lactulose and rifaximin were started to decrease ammonia, and antibiotics (Ampicillin Sodium/Sulbactam 9 g/day) were also started for bronchopneumonia. On the next day following admission, AST, LDH, and CK were still elevated. The plasma myoglobin was 4300 ng/mL and myoglobin in the urine was 73,900 ng/mL. The patient was further complicated with rhabdomyolysis. His respiratory status had been stable; however, it was a critical condition regarding ALF and rhabdomyolysis. Consultation took place with infectious disease specialists, and 1600 mg/d of molnupiravir was started. Transaminases and CK were decreased from the third day and total bilirubin was elevated to 5.5 mg/dL on the fourth day. It finally recovered to the normal level two months later. Renal function was not affected throughout the clinical course. Prothrombin time was recovered by over 40% (below INR 1.5) on the fourth day. The SARS-CoV-2 was finally undetectable on the 26th day. The patient’s hemoglobin recovered to 12.5 g/dL on the 46th day (Figure 3) and the CTR decreased to normal (Figure 2B).

## 3. Discussion

In this case study, we considered a recovered patient with COVID-19 who experienced complications, including ALF and rhabdomyolysis. Notably, this represents the first documented case of the simultaneous occurrence of rhabdomyolysis and ALF associated with SARS-CoV-2 infection.

Several reports have highlighted the occurrence of liver enzyme abnormalities, acute hepatitis, and an ALF associated with SARS-CoV-2 infection [26,27,28,29,30,31,32,33,34,35,36,37,38,39,40,41,42,43]. It has been documented that approximately 76.3% to 82.5% of COVID-19 cases display abnormal liver test results [35,37]. Medetalibeyoglu et al. further demonstrated that elevated levels of AST and ALT, along with an AST/ALT ratio greater than 1, indicated a more severe disease course and increased mortality in COVID-19 patients [26]. However, their study reported maximum AST and ALT values of 421 IU/L and 610 IU/L, respectively. In contrast, our case exhibited significantly higher levels, with an AST/ALT ratio of 2.45. Moreover, Sobotka et al. indicated that severe acute liver injury (ALI) was infrequent, occurring in 0.1% of patients upon admission and 2% during hospitalization, with no ALF observed among a cohort of 1555 patients [30]. The available literature on ALF in the context of COVID-19 remains limited, with only a handful of reports documenting such cases [39,40,41,42,43]. The reports on ALF cases are provided in Table 2, with the main findings summarized.

The case reported by Sarkar et al. is abstract and lacks adequacy in diagnosing ALF [42]. The summary reveals that the median age of affected individuals was 57.5 years old and notable gender distribution was observed, with a male-to-female ratio of 2:1. Among the cases reported, including the present case, two individuals achieved a complete recovery. Two patients died on the ninth day. Weber et al. provided a detailed discussion on the mechanism of liver injury induced by SARS-CoV-2, highlighting the upregulation of the angiotensin-converting enzyme 2 (ACE2) receptor as a contributing factor [41]. ACE2 receptors serve as binding sites for SARS-CoV-2 virions, primarily in the lungs [44]. Furthermore, Diaz suggested that patients who are prescribed ACE inhibitors (ACEIs) and angiotensin receptor blockers (ARBs) may face an elevated risk of experiencing severe disease outcomes in the context of SARS-CoV-2 infections [45]. The present case had never been prescribed ACEIS or ARBs.

Hypoxic hepatitis (HH) has been recognized as a potential precipitating factor for ALF, with an estimated prevalence of 39–70% of HH patients presenting with concurrent heart failure [46]. The primary underlying mechanism for this association primarily involves passive hepatic congestion, resulting from acute heart failure. Other contributing factors include septic shock, respiratory failure, and anemia. An interesting aspect to explore is the examination of COVID-19-related cases of ALF in the context of HH. Notably, in the reviewed cases, transaminase levels peaked within 72 h, consistent with the typical clinical course of HH, except for Cases 2 and 3. This transient pattern suggests that COVID-19 may have triggered HH, leading to the subsequent development of ALF. In the present case, although the patient did not display respiratory failure or septic shock, the presence of underlying heart failure and anemia presented a plausible scenario in which the co-occurrence of COVID-19 may have precipitated HH.

Although direct traumatic injury is the most prevalent cause of rhabdomyolysis, other factors, such as medications, toxins, infections, muscle ischemia, electrolyte and metabolic disorders, genetic disorders, physical exertion or prolonged bed rest, and temperature-induced conditions such as neuroleptic malignant syndrome and malignant hyperthermia, can contribute to the development of rhabdomyolysis [47]. Furthermore, since the 1970s, numerous viruses, including influenza A/B, parainfluenza, rhinovirus, echovirus, coxsackievirus, Epstein–Barr virus, herpes simplex virus, adenovirus, and cytomegalovirus, have been reported to be associated with cases of rhabdomyolysis [48,49,50]. Viruses may affect the musculature directly by invasion or indirectly via immune mechanisms [51,52]. Muscle cells have been found to possess the ACE2 receptor, which serves as the entry pathway for SARS-CoV-2 [53,54]. The second theory postulates that skeletal muscle damage could be caused by the host “cytokine storm”-like immune response [51]. In a systematic review conducted by Hannah et al., various cases of rhabdomyolysis associated with COVID-19, including vaccine-induced cases, were analyzed [22]. Their study revealed that the median age of rhabdomyolysis patients was 50 years. Additionally, 49% of these patients had at least one of the following conditions: hypertension, diabetes mellitus, or obesity. It was also observed that 77% of the patients were male. Among the cases, 28% required intravenous hemofiltration and 36% underwent mechanical ventilation. The mortality rate among COVID-19 patients with rhabdomyolysis is higher than the previously estimated in-patient mortality rate for COVID-19 (30% vs. 12%). Conversely, Bawor et al. indicated that the prognosis is favorable among patients with normal renal function, such as the current case [23]. In the systematic review, cardiac muscle involvement was also found in 7%. In the present case, the potential diagnosis of myocarditis was excluded, and COVID-19-related heart failure was not considered [22].

Among the reports, including a systematic review that provided laboratory data for reference, none documented cases that were complicated by both ALF and rhabdomyolysis in association with COVID-19 [29,30,31,32,33,34,35,36,37,38,39,40,41,42,43,44,45,51]. However, the information presented in these reports shows resemblances in the underlying factors between ALF and rhabdomyolysis. Both conditions appear to be more prevalent in older males with underlying diseases, primarily hypertension.

In the present case, molnupiravir was used. Molnupiravir is an oral prodrug of beta-D-N4-hydroxycytidine (NHC), which was approved by the Food and Drug Administration (FDA) on 23 December 2021 and by the Ministry of Health, Labor and Welfare in Japan on 24 December 2021. Molnupiravir is recommended for treating non-hospitalized adults with mild to moderate COVID-19 within five days of symptom onset and at high risk of progressing to severe disease and is expected to be active against the Omicron variant and its subvariants [55]. The current case was considered to have a high risk of progressing to severe disease, and the Omicron variant was also anticipated. Furthermore, when compared to remdesivir and favipiravir, molnupiravir demonstrates a relatively safe profile without any observed liver toxicity [56]. When considering COVID-19 patients who exhibit multiorgan failure, molnupiravir is a favorable treatment option.

This report has a limitation pertaining to the inability to perform a liver biopsy on the patient, due to coagulopathy, resulting in the unavailability of histological information. However, in the context of severe liver failure, histological studies are not deemed obligatory. Ultimately, the clinical progression of the patient did not exhibit indications of other underlying diseases, thus facilitating a comprehensive assessment of the pathophysiology.

## 4. Conclusions

This case report highlights the occurrence of ALF and rhabdomyolysis in a patient with COVID-19. Furthermore, it suggests that COVID-19 can trigger severe illnesses, irrespective of the absence of respiratory failure.

## Figures and Tables

**Figure 1 viruses-15-01445-f001:**
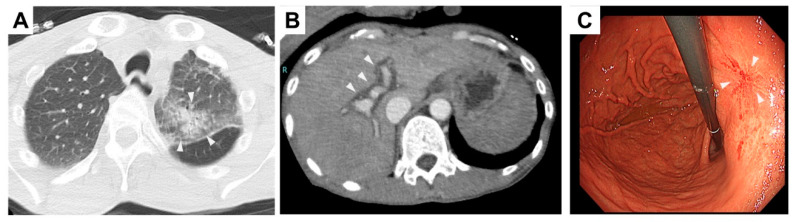
Imaging findings. (**A**) Chest CT showed bronchopneumonia in the left upper lobe of the lung (arrowheads). (**B**) Abdominal enhanced CT showed a periportal collar sign (arrowheads), implying acute hepatitis. (**C**) Gastroduodenal endoscopic examination revealed a scar of peptic ulcer at the lesser curvature of the stomach (arrowheads).

**Figure 2 viruses-15-01445-f002:**
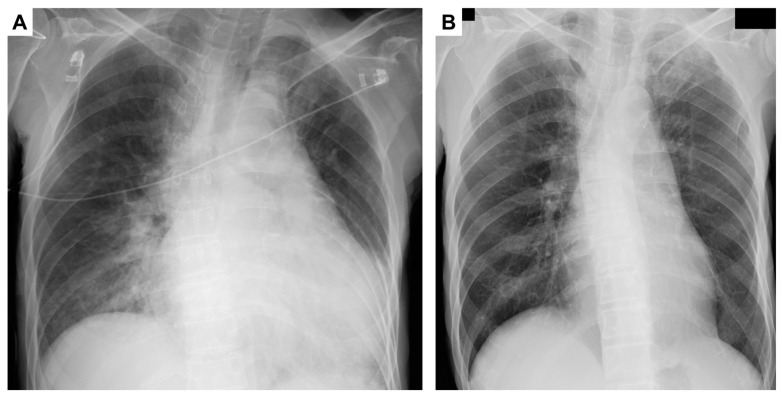
Chest radiograph findings: (**A**) CTR was 67.6% on the day of admission and (**B**) on the 28th day showed improvement in the CTR from 67.6% to 49.9%. CTR, cardiothoracic ratio.

**Figure 3 viruses-15-01445-f003:**
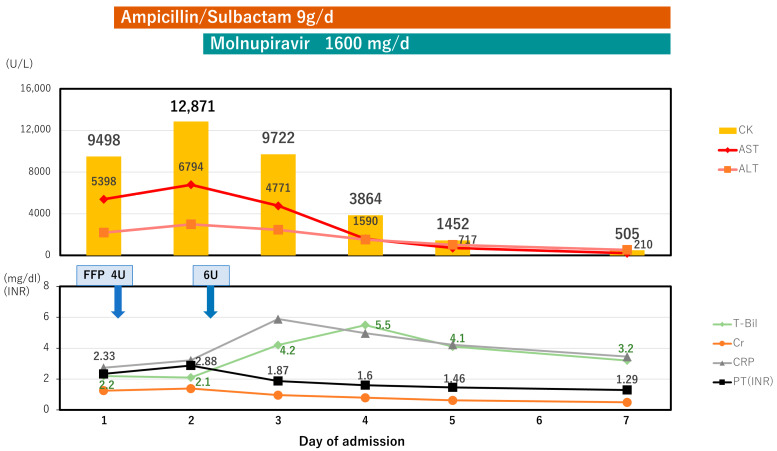
Clinical course. Antibiotics (Ampicillin Sodium/Sulbactam 9 g/day) was started for bronchopneumonia. A total of 10 units of FFP were administered to address acute liver failure. Further elevations in AST, LDH, and CK levels were observed on the second day. Elevated levels of blood myoglobin and urinary myoglobin indicated the presence of rhabdomyolysis, leading to the initiation of monnupilavir treatment. By the third day, liver enzyme levels and CK began to decrease. Although there was a tendency for bilirubin to increase, it subsequently decreased by the fifth day. FFP, fresh frozen plasma; AST, aspartate aminotransferase; LDH, lactate dehydrogenase; Cre, creatinine; CK, creatine kinase; CRP, c-reactive protein; T-bil, total bilirubin; PT, prothrombin time.

**Table 1 viruses-15-01445-t001:** Laboratory data on admission.

CBC				Serology			
WBC	12.2	(3.3–8.6)	×10^3^/µL	anti-EBV VCA IgG	1:320	(+)	
Neut	91.4	(38.5–80.5)	%	anti-EBV VCA IgM	<1:10	(−)	
Lymph	4.6	(16.5–49.5)	%	anti-CMV IgG	31.4	(+)	
Mon	4	(2.0–10.0)	%	anti-CMV IgM	0.32	(−)	
Eos	0	(0.0–8.5)	%	anti-HSV IgG	31.4	(+)	
Bas	0	(0.0–2.5)	%	anti-HSV IgM	0.32	(−)	
RBC	2.46	(3.86–4.92)	×10^4/^mL	HBs Ag	(−)		
Hb	3.4	(11.6–14.8)	g/dL	HBs Ab	(−)		
Hct	15.4	(35.1–44.4)	%	HCV Ab	(−)		
MCV	62.6	(83.6–98.2)	fL	IgM-HA Ab	<1:40		
MCH	13.8	(27.5–33.2)	%	IgA-HEV Ab	(−)		
MCHC	22.1	(31.7–35.3)	%	ANA	<1:40		
Platelet	18.9	(15.8–34.8)	×10^4^/µL	AMA-M2	<1.5	(−)	
Biochemistry				Ig G	1493	(861–1747)	mg/dL
Total protein	7.0	(6.6–8.1)	g/dL	Ig G4	57	(11–121)	mg/dL
Albumin	3.9	(4.1–5.1)	g/dL	Ig M	37	(50–269)	mg/dL
Total bilirubin	2.2	(0.4–1.5)	mg/dL	SARS-CoV-2	(+)		
Direct bilirubin	0.9	(≤0.4)	mg/dL	N (Ct)	25.1		
AST	5398	(13–30)	U/L	N (copies)	666,135.0		
ALT	2197	(7–23)	U/L	N2 (Ct)	20.3		
ALP (IFCC)	99	(38–113)	U/L	N2 (copies)	2,420,950.0		
GGT	55	(9–32)	U/L	Coagulation			
LDH	4636	(124–222)	U/L	PT (%)	26.2	(70–130)	%
CK	9498	(41–153)	U/L	PT(INR)	2.33	(0.80–1.27)	
BUN	31.3	(8–20)	mg/dL				
Creatinine	1.25	(0.46–0.79)	mg/dL				
CRP	2.73	(<0.15)	mg/dL	Plasma myoglobin	4366	(≤154.9)	ng/mL
NH3	159	(12–66)	µg/dL	Urine myoglobin	73,900	(≤2.0)	ng/mL

AST, aspartate aminotransferase; ALT, alanine aminotransferase; ALP, alkaline phosphatase; GGT, γ-glutamyl transpeptidase; ANA, anti-nuclear antigen; AMA, anti-mitochondrial antigen, Baso, basophil; BUN, blood urea nitrogen; CBC, complete blood count; CMV, cytomegalovirus. CRP, c-reactive protein; Ct, Cycle Threshold; EBV, Epstein–Barr virus; EBNA, Epstein–Barr virus nuclear antigen; Eo, eosinophil; HA, hepatitis A; HB, hepatitis B; Hct, hematocrit; HCV, hepatitis C virus; HEV, hepatitis E virus; HSV, herpes simplex virus; IFCC, International Federation of Clinical Chemistry and Laboratory Medicine; Ig, immunoglobulin; INR, international normalized ratio; LDH, lactate dehydrogenase; Lymph, lymphocyte; Mon, monocyte; Neuto, neutrophil; MCV, mean corpuscular volume; MCH, mean corpuscular hemoglobin; MCHC, mean corpuscular hemoglobin concentration; TP, total protein; PT, prothrombin time; SARS-CoV-2, severe acute respiratory syndrome coronavirus 2; VCA, virus capsid antigen; WBC, white blood cell. (Reference values).

**Table 2 viruses-15-01445-t002:** Published reports of acute liver failure associated with COVID-19.

Case#	Age	Sex	Underlying Diseases	Highest AST (IU/L)	Highest ALT (IU/L)	T-bil (mg/dL)	PT-INR	CK	CRP (mg/dL)	Prognosis	Ref.
1	35	F	SLE	4202 (day 3)	5524 (day 3)	10.5 (day 3)	4.9	n/a	6.68	recovered	[39]
2	80	M	DM, HT, HLP, CAD, asthma,	>7000 * (day 5)	3737 (day 5)	8.4 (day 8)	8.94 (day 8)	n/a	n/a	died (day 9)	[40]
3	65	M	HT	746 (day 14)	467 (day 14)	22.2 (day 20)	2–3 ** (day 20)	n/a	n/a	n/a	[41]
4	53	M	CM	4735 (day 2)	1988 (day 2)	n/a	n/a	n/a	n/a	n/a	[42]
5	49	F	AD, HT, DA	950 (day 1)	1375 (day 1)	21.2 (day 6)	15.5 (day 4)	n/a	n/a	died (day 9)	[43]
6	62	M	PU, HF	6798 (day 2)	2987 (day 2)	6.0 (day 19)	2.88 (day 2)	12,871 (day 2)	5.89	recovered	Presentcase
**summary**	**Median** **57.5**	**66.7%** **Male**		**Median** **4469**	**Median** **2488**	**Median** **9.45**	**Median** **4.9**				

AD, aortic dissection; AST, aspartate aminotransferase; ALF, acute liver failure; ALT, alanine aminotransferase; BPH, benign prostate hypertrophy; CM, cardiomyopathy; CK, creatine kinase; CKD, chronic kidney disease; CRP, c-reactive protein; DA, drug abuse; DM, diabetes mellitus; HF, heart failure; HTN, hypertension; IGT, impaired glucose tolerance; INR, international normalized ratio; LDH, lactate dehydrogenase; n/a, not available; OB, obesity; OSAS, obstructive sleep apnea syndrome; PT, prothrombin time; PU, peptic ulcer; SLE, systemic lupus erythematosus; T-bil, total bilirubin; n/a, not available.* calculate the median as 7000. ** Values are not indicated on the graph; therefore, calculate the median as 2.5.

## Data Availability

Not applicable.

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
