# Peer review of "COVID-19-Triggered Acute Liver Failure and Rhabdomyolysis: A Case Report and Review of the Literature"

_viruses, 2023, doi:10.3390/v15071445_

Round 1

Reviewer 1 Report

This manuscript by Matsuki et al. has two major findings:

1- Report of a human clinical case with COVID-19 followed by acute liver failure (ALF) and rhabdomyolysis.

2- Literature review on cases of ALF triggered by SARS-CoV-2 infection.

The authors' study is relevant in the context of COVID-19 pandemic. However as highlighted below, overall, the introduction (background information), methodology and Discussion would benefit from a more coherent structure and concise explanation of the researches.

1. The authors performed a review of AFL associated with SARS-CoV-2 infection but did not conduct a review of rhabdomyolysis associated with SARS-CoV-2 infection. Wasn't this done just because there is already a systematic review?

2. Title: it is wrong to write COVID-19 infection. Please change to Severe COVID-19 Complicated...

3. Abstract: It does not say COVID-19 infection but SARS-CoV-2 infection. Please take care to make the necessary adjustments throughout the manuscript.

4. Introduction: I see it as important that the authors already mention the clinical-laboratory parameters of a diagnosis of ALF and rhabdomyolysis.

5. Table 1: add reference values of clinical and laboratory parameters.

6. Figures 1-2: Place arrowheads on figures to highlight key clinical findings.

7. Table 2: For the present case report, the age is 60 years old, however previously it was said that the male was 62 years old.

8. For reports of ALF associated with SARS-CoV-2 infection there are other studies that were not included in the analysis.

9. It is important for the authors to discuss the limitations of their study.

Author Response

Q1. The authors performed a review of AFL associated with SARS-CoV-2 infection but did not conduct a review of rhabdomyolysis associated with SARS-CoV-2 infection. Wasn't this done just because there is already a systematic review?

A1: Thank you for your question. As you pointed out, a systematic review already exists. We thoroughly read these available reports on rhabdomyolysis, and no reports showed co-occurrence with ALF. We mentioned in the manuscript“Among the reports, including a systematic review that provided laboratory data for reference, none of them documented cases complicated by both ALF and rhabdomyolysis in association with COVID-19 [29-45, 51].”

Q2. Title: it is wrong to write COVID-19 infection. Please change to Severe COVID-19 Complicated...

Q3. Abstract: It does not say COVID-19 infection but SARS-CoV-2 infection. Please take care to make the necessary adjustments throughout the manuscript.

A2,3: We are very sorry for such a confusing expression. We changed the title to COVID-19 Triggered…”. We also made changes from COVID-19 infection to SARS-CoV-2 infection.

Q4. Introduction: I see it as important that the authors already mention the clinical-laboratory parameters of a diagnosis of ALF and rhabdomyolysis.

A4. We added, “In Japanese diagnostic criteria, patients showing prothrombin time values of 40% or less of the standardized values or international normalized ratios (INR) of 1.5 or more caused by severe liver damage within 8 weeks of the onset of the symptoms are diagnosed as having ALF [24]. Rhabdomyolysis is diagnosed by elevations in serum creatine kinase (CK), and while there is no established serum level cut-off, many clinicians use five times the upper limit of normal (~1000 U/l)[25].” at the end of the introduction.

Q5. Table 1: add reference values of clinical and laboratory parameters.

A5: We added the reference values in Table1

Q6. Figures 1-2: Place arrowheads on figures to highlight key clinical findings.

A6: We added arrowheads in the Figure 1.

Q7. Table 2: For the present case report, the age is 60 years old, however previously it was said that the male was 62 years old.

A7: We fixed the age from 60 to 62 years old in Table 2.

Q8. For reports of ALF associated with SARS-CoV-2 infection there are other studies that were not included in the analysis.

A8: An additional report documenting ALF was identified and included as a reference [43] in Table 2. It is worth noting that there are several reports discussing ALF cases potentially triggered by COVID-19 vaccination or acute-on-chronic liver failure; however, these reports were excluded from the present analysis because of the pathophysiological differences.

Q9. It is important for the authors to discuss the limitations of their study.

A9: We added the limitations as “This report has a limitation pertaining to the inability to perform a liver biopsy on the patient due to coagulopathy, resulting in the unavailability of histological information. However, in the context of severe liver failure, histological studies are not deemed obligatory. Ultimately, the clinical progression of the patient did not exhibit indications of other underlying diseases, thus facilitating a comprehensive assessment of the pathophysiology.” at the end of the discussion.

Reviewer 2 Report

In this manuscript, Matsuki et al. showed a case of acute liver failure (ALF) and rhabdomyolysis associated with severe COVID-19 infection. This case report is very rare and novel. However, there are some issues that should be addressed.

Major

1.     This case was complicated by ALF and rhabdomyolysis due to severe COVID-19, but also severe anemia and heart failure. In Figure 3, transaminases improved quickly. The possibility of hepatic and muscular ischemia should be considered.

2.     You stated that the cause of the heart failure was severe anemia, but the rationale and differential diagnosis are unclear. Have you differentiated myocarditis, acute coronary syndrome, or takotsubo cardiomyopathy? In the discussion, you stated that COVID-19 might influence heart failure. You should also describe the ECG findings.

3.     Figure 2 showed improvement in cardiac enlargement. Was the patient treated for heart failure?

4.     Peptic ulcer was observed in this case. Was there a history of black stools?

5.     Regardless of rhabdomyolysis, this case had no tenderness over the major muscle group. Were antipyretic analgesics used in this case?

 Minor

1.   Please check the Glasgow Coma Scale on admission (E4V4M6?).

Author Response

Major

Q1. This case was complicated by ALF and rhabdomyolysis due to severe COVID-19, but also severe anemia and heart failure. In Figure 3, transaminases improved quickly. The possibility of hepatic and muscular ischemia should be considered.

A1: Thank you for your questions. We added in the case presentation as “A shock status or a hypotensive event was not documented throughout the transference,”, and “Contrast-enhanced CT imaging revealed the absence of ischemic alterations in both the hepatic and muscular regions.” Moreover, as also Reviewer 3 indicated, we discussed hypoxic hepatitis as a possible cause of ALF as “Hypoxic hepatitis (HH) has been recognized as a potential precipitating factor for liver failure, with an estimated prevalence of 39-70% of HH patients presenting with concurrent heart failure [46]. The primary underlying mechanism for this association primarily involves passive hepatic congestion resulting from acute heart failure. Other contributing factors include septic shock, respiratory failure, and anemia. An interesting aspect to explore is the examination of COVID-19-related cases of ALF in the context of HH. Notably, in the reviewed cases, transaminase levels peaked within 72 hours, consistent with the typical clinical course of HH, except for Cases 2 and 3. This transient pattern suggests that COVID-19 may have triggered HH, leading to the subsequent development of acute liver failure. In the present case, although the patient did not display respiratory failure or septic shock, the presence of underlying heart failure and anemia presents a plausible scenario in which the co-occurrence of COVID-19 may have precipitated HH.” in the discussion.

Q2. You stated that the cause of the heart failure was severe anemia, but the rationale and differential diagnosis are unclear. Have you differentiated myocarditis, acute coronary syndrome, or takotsubo cardiomyopathy? In the discussion, you stated that COVID-19 might influence heart failure. You should also describe the ECG findings.

A2: We added, Electrocardiographic monitoring findings did not exhibit abnormalities such as ST elevation or negative T waves. Echocardiography demonstrated an enlarged left ventricle (LVDd 53.5mm, LVDs 42.0mm) and a ventricular ejection fraction (vEF) ranging from 20% to 25%. A severe reduction in wall motion was observed throughout the myocardium. The characteristic features of "takotsubo cardiomyopathy," including apical ballooning, symmetrical regional abnormalities in the form of a circumferential pattern, and left ventricular outflow tract obstruction, were absent in the observed cases. Notably, there were no indications of cardiac valvular disease. Moreover, the CK-MB values (64 U/L) displayed a relatively modest elevation compared to the high CK values, which deviated from the expected patterns typically observed in cases of myocarditis. These observations significantly reduced the likelihood of myocarditis as a plausible diagnosis. Furthermore, despite calcification in the coronary arteries, the absence of wall thinning suggested that acute coronary syndrome was an unlikely explanation. While the potential for type 2 myocardial infarction resulting from anemia was considered, the cardiologists ultimately determined that the patient's heart failure was secondary to stress-induced myocardial injury, particularly in severe conditions such as anemia, liver failure, and rhabdomyolysis.” in the case presentation. We also changed the sentence to “In the present case, the potential diagnosis of myocarditis was excluded, and COVID-19-related heart failure was not considered [22]” in the discussion.

Q3. Figure 2 showed improvement in cardiac enlargement. Was the patient treated for heart failure?

A3: No targeted treatment for heart failure was administered; instead, the patient was just treated for anemia with a red blood cell transfusion. We added, “Consequently, no specific therapeutic interventions to address heart failure were administered, and the patient received red blood cell transfusions to manage the underlying anemia.”

Q4. Peptic ulcer was observed in this case. Was there a history of black stools?

A4: The patient had no history of black stool before admission. We added, “However, the patient did not report any prior melena.” in the case presentation.

Q5. Regardless of rhabdomyolysis, this case had no tenderness over the major muscle group. Were antipyretic analgesics used in this case?

A5: Antipyretic analgesics were not used in this case on admission. We added the sentence “He did not take any antipyretic analgesics” in the case presentation. We speculate that the patient was very thin; therefore, originally, the muscle volume was small. It may be the reason why the patient had no remarkable tenderness.

 Minor

.   Please check the Glasgow Coma Scale on admission (E4V4M6).

A1: We are very sorry for such a mistake. We fixed it as E4V4M6.

Reviewer 3 Report

The Case Report by Matsuki Y. et al. describes Hypoxemic Hepatitis in a patient with severe heart failure, severe anemia, and severe pneumonia (by COVID) as the origin of liver failure and rhabdomyolysis. Therefore, COVID can be one important variable to produce hypoxemia but not the only (Journal of Clinical and Translational Hepatology 2016 vol. 4 |263–268)

Authors should make important changes in the title, diagnosis, and review focus on the causes of hypoxemic hepatitis and its management to improve their Case Report, avoid wrong conclusions, and be accepted in viruses

Author Response

1.The Case Report by Matsuki Y. et al. describes Hypoxemic Hepatitis in a patient with severe heart failure, severe anemia, and severe pneumonia (by COVID) as the origin of liver failure and rhabdomyolysis. Therefore, COVID can be one important variable to produce hypoxemia but not the only (Journal of Clinical and Translational Hepatology 2016 vol. 4 |263–268) Authors should make important changes in the title, diagnosis, and review focus on the causes of hypoxemic hepatitis and its management to improve their Case Report, avoid wrong conclusions, and be accepted in viruses

Answer:

We appreciate for suggesting a very important point of view. This case was not complicated by severe pneumonia and respiratory failure; however, as Reviewer 3 indicated that COVID-19 might exacerbate severe anemia and secondary heart failure and induce hypoxic hepatitis. We changed the title to “COVID-19 Triggered Acute Liver Failure and Rhabdomyolysis: A Case Report and Review of the Literature”. We also added, “Hypoxic hepatitis (HH) has been recognized as a potential precipitating factor for liver failure, with an estimated prevalence of 39-70% of HH patients presenting with concurrent heart failure [46]. The primary underlying mechanism for this association primarily involves passive hepatic congestion resulting from acute heart failure. Other contributing factors include septic shock, respiratory failure, and anemia. An interesting aspect to explore is the examination of COVID-19-related cases of acute liver failure in the context of HH. Notably, in the reviewed cases, transaminase levels peaked within 72 hours, consistent with the typical clinical course of HH, except for Cases 2 and 3. This transient pattern suggests that COVID-19 may have triggered HH, leading to the subsequent development of acute liver failure. In the present case, although the patient did not display respiratory failure or septic shock, the presence of underlying heart failure and anemia presents a plausible scenario in which the co-occurrence of COVID-19 may have precipitated HH.”in the discussion.

Round 2

Reviewer 1 Report

In accordance with the reviewers' recommendations, the manuscript was improved. Therefore, I agree with the acceptance of the manuscript in its present form.

Reviewer 2 Report

The authors adequately addressed the reviewers' questions.

Reviewer 3 Report

The authors have made modifications to the original manuscript based on the reviewers’ comments and advice improving the quality of their study. 

Now, the manuscript is suitable for publication in viruses